# A Paleocene penguin from New Zealand substantiates multiple origins of gigantism in fossil Sphenisciformes

Gerald Mayr[1], R. Paul Scofield[2], Vanesa L. De Pietri[2] & Alan J.D. Tennyson[3]

One of the notable features of penguin evolution is the occurrence of very large species in the early Cenozoic, whose body size greatly exceeded that of the largest extant penguins. Here we describe a new giant species from the late Paleocene of New Zealand that documents the very early evolution of large body size in penguins. *Kumimanu biceae*, n. gen. et sp. is larger than all other fossil penguins that have substantial skeletal portions preserved. Several plesiomorphic features place the new species outside a clade including all post-Paleocene giant penguins. It is phylogenetically separated from giant Eocene and Oligocene penguin species by various smaller taxa, which indicates multiple origins of giant size in penguin evolution. That a penguin rivaling the largest previously known species existed in the Paleocene suggests that gigantism in penguins arose shortly after these birds became flightless divers. Our study therefore strengthens previous suggestions that the absence of very large penguins today is likely due to the Oligo-Miocene radiation of marine mammals.

---

[1] Senckenberg Research Institute and Natural History Museum Frankfurt, Ornithological Section, Senckenberganlage 25, D-60325 Frankfurt am Main, Germany. [2] Canterbury Museum, Rolleston Avenue, Christchurch 8013, New Zealand. [3] Museum of New Zealand Te Papa Tongarewa, PO Box 467, Wellington 6140, New Zealand. Correspondence and requests for materials should be addressed to G.M. (email: Gerald.Mayr@senckenberg.de)

New Zealand has yielded several fossils of Paleocene Sphenisciformes, which shed considerable light on the early evolution of penguins. All of the described specimens come from exposures of the Waipara Greensand in the Canterbury region and the two named species, *Waimanu manneringi* and *W. tuatahi*, are the oldest and phylogenetically most basal Sphenisciformes reported so far[1–3].

Recently, remains of a very large penguin have also been found in the Waipara Greensand[4]. These fossils, an incomplete tarsometatarsus and associated pedal phalanges, belong to an unnamed species that is phylogenetically closer to the crown group (the clade including the extant species) than *Waimanu*. A definitive taxonomic assignment of the fragmentary fossils is, however, not possible and this is also true for *Crossvallia unienwillia*, an equally large stem group penguin from the late Paleocene of Antarctica[5, 6].

Gigantism, that is, the evolution of a size exceeding that of the extant Emperor Penguin (*Aptenodytes forsteri*), is much better

**Fig. 1** Wing and pectoral girdle bones of the new giant penguin. **a** *K. biceae* n. gen. et sp. (holotype, NMNZ S.45877) from the Paleocene of New Zealand, partially prepared concretion with all bones in situ. **b** *K. biceae* (holotype), right coracoid in dorsal view (dotted lines indicate reconstructed outline of bone). **c** Left coracoid of *Waimanu tuatahi* from the late Paleocene of New Zealand (CM zfa 34; specimen mirrored to ease comparisons). **d–f** Fragmentary proximal end of the left ulna of *K. biceae* in (**d**) dorsal, (**e**) ventral, and (**f**) proximal view. **g, h** Left ulna of an undescribed new sphenisciform from the Waipara Greensand (CM 2016.6.1) in (**g**) ventral and (**h**) proximal view; the dashed line in **g** indicates the portion of the bone preserved in the *K. biceae* holotype. **i** CT image of cranial surface of partial left humerus. **j** Exposed caudal surface of the bone, surrounding bones and matrix were digitally brightened. **k, l** CT images of caudal humerus surface with (**k**) minimum and (**l**) maximum length estimates based on the reconstructed outline of the bone (dotted lines). **m** Left humerus of *Crossvallia unienwillia* from the late Paleocene of Antarctica (holotype, MLP 00-I-10-1), which is one of the largest previously known Paleocene penguin species. **n** Left humerus of *Pachydyptes ponderous* from the late Eocene of New Zealand (holotype, NMNZ OR.1450), which was previously considered one of the largest fossil penguins. Abbreviations: cor, coracoid; dcp, dorsal cotylar process; fem, femur; fpt, fossa pneumotricipitalis; hum, humerus; olc, olecranon; ppc, procoracoid process; scc, scapular cotyla; sup, attachment scar for supracoracoideus muscle; vct, ventral cotyla. Scale bars equal 50 mm; same scale for **b** and **c**, **f** and **h**, and **i-l**, respectively

**Table 1 Selected bone dimensions (in mm) of the new species and other Sphenisciformes**

| | Humerus, length | Humerus, maximum width of proximal end | Coracoid, length | Femur, length | Tibiotarsus, distal width |
|---|---|---|---|---|---|
| *Kumimanu biceae* n. gen. et sp. | ~185–228 | 70 | ~224 | 161 | ~48 [reconstr.] |
| *Palaeeudyptes klekowskii* | 150.4–158[10] | 44.7[a] | — | 134.4[b] | 34.7[10] |
| cf. *Palaeeudyptes klekowskii* (MLP 12-I-20-288) | ~259.2[14] | — | — | — | ~53.5[14,c] |
| *Palaeeudyptes gunnari* | 142.2–144.1[10] | 42.6[d] | ~146.2[10] | 123.6[10] | 28.8–30.0[10] |
| *Palaeeudyptes marplesi* | — | — | — | 142.9[10] | — |
| *Kairuku grebneffi* (holotype) | 176.6[8] | 55.5 | 191.5[8] | 143.9[8] | ~40.6 |
| *Kairuku waitaki* (holotype) | — | ~48.1 | 187.1[8] | 127.3[8] | ~42.8 |
| *Pachydyptes ponderosus* | 179.6 | 68.2 | ~224 | — | — |
| "*Pachydyptes*" *simpsoni* | — | — | ~171[17] | — | — |
| *Inkayaku paracasensis* | 161.6[19] | — | 173.0[19] | 145.9[19] | — |
| *Icadyptes salasi* | 167.0[32] | 61.7[32] | — | — | — |
| *Anthropornis nordenskjoeldi* | 152.2[10]–181[e] | ~57[e] | — | — | 44.7[10] |
| *Anthropornis grandis* | 103.2[10] | — | ~197.1[10] | — | — |
| *Archaeospheniscus wimani* | — | — | — | 124.6[10] | — |
| *Crossvallia unienwillia* | 170.9[5] | ~53 | — | ~135[5] | 35.7[5] |
| unnamed giant Waipara penguin (CM 2016.158.1) | — | — | — | — | ~40.5[4,c] |
| *Aptenodytes forsteri* | 121.0–124.0[33] | 39.2–40.0[33] | 166.0–182.0[33] | 112.2–117.0[33] | 28.0–29.7[33] |

Unreferenced measurements were taken by the authors
[a]specimen MLP 11-2-20-07
[b]specimen MLP 12-1-20-289
[c]size estimate based on width of proximal end of tarsometatarsus
[d]specimen MLP 96-1-6-13
[e]specimen MLP 83-1-1-190

documented in post-Paleocene penguins, and the Eocene *Anthropornis nordenskjoeldi* and *Pachydyptes ponderosus* were for a long time considered to be the largest known penguin species[7, 8]. *Pachydyptes ponderosus*, from the late Eocene of New Zealand, is known only from a few wing and pectoral girdle bones[7]; however, numerous isolated skeletal elements as well as a few partial skeletons have been reported for *A. nordenskjoeldi*, from the late Eocene and early Oligocene of Antarctica[9, 10]. Recently, it was hypothesised that the well-preserved *Kairuku grebneffi* from the late Oligocene of New Zealand may have been taller, although less massive than *P. ponderosus*; for the largest individual of *K. grebneffi*, a total body length of about 1.5 m was estimated[8]. Based on isolated limb bones, lengths of 1.6 and 1.5 m were also calculated for *Anthropornis nordenskjoeldi* and the very large *Palaeeudyptes klekowskii* from the Eocene and Oligocene of Antarctica[11]. While partial skeletons of *P. klekowskii* indicate a somewhat shorter body length of about 1.4 m[12, 13], a recently described humerus fragment and a tarsometatarsus may come from individuals with an estimated length of about 2.0 m[14]. A large size is reached by other *Palaeeudyptes* species from the Eocene and Oligocene of Antarctica and New Zealand[2, 15, 16], and further, very large Sphenisciformes occurred in the late Eocene of Australia[17] and the late Eocene of Peru[18, 19].

Some authors assumed that penguins achieved a giant size multiple times[6], but the giant taxa *Anthropornis*, *Palaeeudyptes*, *Kairuku*, *Icadyptes*, and *Inkayacu* were recovered as parts of subsequently branching clades and it was therefore considered more likely that extremely large size evolved only once[18]. Definitive conclusions about size evolution in fossil Sphenisciformes are, however, impeded by the fact that even in more recent analyses the exact interrelationships between giant sphenisciform taxa are poorly resolved[8, 19, 20].

Here we report a partial skeleton of a giant stem penguin from the Paleocene Moeraki Formation at Hampden Beach in the Otago region of New Zealand, some 300 km southwest of the exposures of the Waipara Greensand in the Canterbury region. A few fragmentary bird remains from the Moeraki Formation were previously mentioned[21] and the age of the Moeraki Formation has been constrained to the late Paleocene based on foraminiferal biostratigraphy[22, 23]. The new fossil is one of the oldest giant penguins found so far and is clearly outside a clade including the giant Eocene and Oligocene Sphenisciformes, substantiating multiple origins of gigantism in fossil penguins.

## Results
**Systematic paleontology.**

Aves Linnaeus, 1758
Sphenisciformes Sharpe, 1891
*Kumimanu biceae*, n. gen. et sp.

**Holotype**. NMNZ S.45877: partial skeleton of a single individual including cranial end of left scapula, incomplete right coracoid, cranialmost portion of sternum, partial left humerus, incomplete proximal end of left ulna, right femur, right tibiotarsus lacking proximal end, partial synsacrum, three vertebrae, and various bone fragments.
**Etymology**. From kumi (Maori), a large mythological monster, and manu (Maori), bird. The species epithet honors Beatrice ("Bice") A. Tennyson, the mother of AJDT, who fostered his interest in natural history (pronounced "bee-chee-ae").
**Type locality and horizon**. Hampden Beach, Otago, New Zealand (NZ Fossil Record Number J42/f0956; precise locality information is recorded at NMNZ); Moeraki Formation, late Paleocene (late Teurian, local stratigraphic level NZP5[22], which has an absolute age of 55.5.-59.5 million years[23]; a matrix sample taken from the fossil (GNS Science sample L29126) contained a specimen of the dinoflagellate *Palaeocystodinium australinum* and an unnamed dino-flagellate taxon that support a Teurian age for this sample; C. Clowes, pers. comm.).

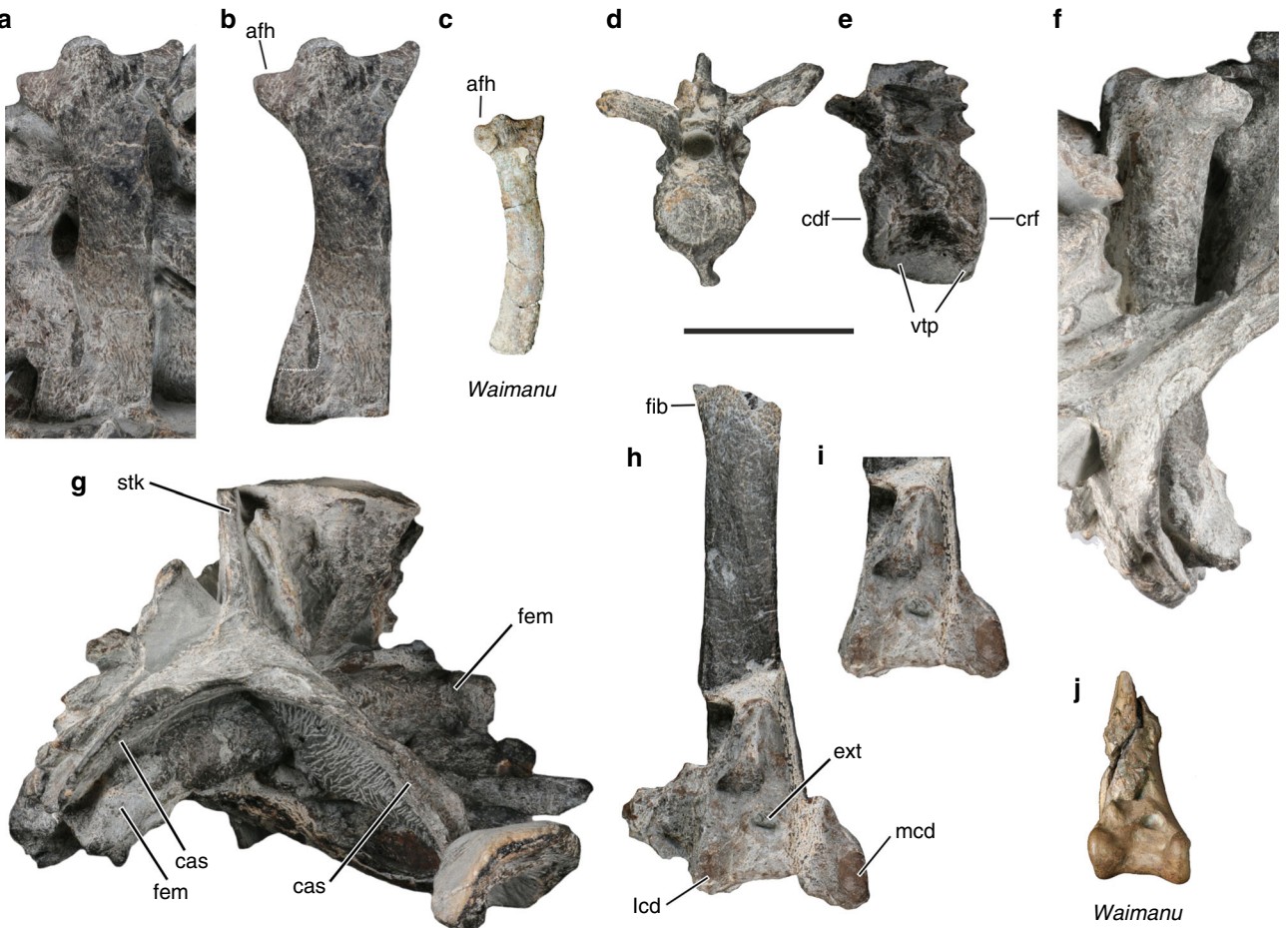

**Fig. 2** Further bones of the giant Paleocene penguin from New Zealand. **a** *K. biceae* n. gen. et sp. (holotype, NMNZ S.45877), cranial portion of left scapula; in **b** the surrounding matrix and bones were digitally removed; the dotted line demarks an overlying bone fragment. **c** Left scapula of *Waimanu tuatahi* from the late Paleocene of New Zealand (CM zfa 34). **d**, **e** Thoracic vertebra of *K. biceae* in (**d**) caudal and (**e**) right lateral view. **f** Right femur of *K. biceae* in craniomedial view. **g** *K. biceae*, sternum in cranial view. **h** Right tibiotarsus in cranial view. **i** Digitally reconstructed distal end of tibiotarsus, in which the medial condyle was brought into its presumed original position and a piece of adhering bone fragment and matrix were removed. **j** Distal end of right tibiotarsus of *Waimanu manneringi* (holotype, CM zfa 35). afh, articulation facet of humerus; cas, coracoidal articulation sulcus; cdf, caudal articulation facet; crf, cranial articulation facet; ext, extensor sulcus; fem, femur; fib, fibular crest; lcd, lateral condyle; mcd, medial condyle; stk, sternal keel; vtp, ventral process. Scale bars equal 50 mm

**Diagnosis**. A very large-sized sphenisciform species, which is characterized by proximodistally low and widely spaced condyles of the tibiotarsus. Distinguished from the late Paleocene *Crossvallia* and all post-Paleocene Sphenisciformes of which humeri are known in the dorsoventrally narrower humerus shaft, with ratio of maximum width of proximal end of humerus to minimum width of shaft being 2.4 (less than this value in *Crossvallia* and all post-Paleocene Sphenisciformes of which the humerus is known). Distinguished from *Waimanu tuatahi* in having the bicipital crest of humerus not forming a distally directed bulge. Distinguished from *Waimanu manneringi* (the humerus of which is unknown) in having the tibiotarsus with proximodistally lower and more widely spaced condyles.

**Dimensions (in mm)**. Coracoid, maximum length as preserved, 141.8; humerus, maximum length as preserved, 149.6; dorsoventral width of proximal end, 75.0; Tibiotarsus, length from distal margin of condylus lateralis to preserved proximal end, 151.3; width of distal end across condyles as preserved, 51.9; estimated actual width of distal end, ~48. Femur, maximum length along longitudinal axis, 161 mm.

**Description and comparisons**. The bones were densely packed in a very hard concretion (Fig. 1a). The humerus (Fig. 1i–l) lacks most of the caput humeri and the distal end, with the latter having already been broken and lost before the bone was embedded in sediment. The attachment scar for the supracoracoideus muscle is slightly raised and dorsoventrally narrower than in *Pachydyptes* (Fig. 1n), *Anthropornis*, and *Icadyptes*. The fossa pneumotricipitalis is single and deeply excavated. The humerus shaft is proportionally narrower than in all post-Paleocene Sphenisciformes, and this is especially true if the bone is compared to the humeri of the giant taxa *Pachydyptes*, *Anthropornis*, and *Icadyptes*. An estimation of the total length of the humerus of *Kumimanu biceae* depends on the assumed proportions. If the proximal end had a similar relative size to that of *Waimanu tuatahi*, less than two thirds of the bone would be preserved and the estimated total length would have been about 228 mm (Fig. 1l). Our most conservative length estimate, by contrast, is based on comparisons with the short (albeit much stouter) humerus of *Anthropornis*—in this case only a small portion of the distal end would be missing and the humerus of *Kumimanu biceae* would have had a length of about 185 mm (Fig. 1k). If the bone was indeed that short, it would have had a proportionally much wider proximal end than the humeri of

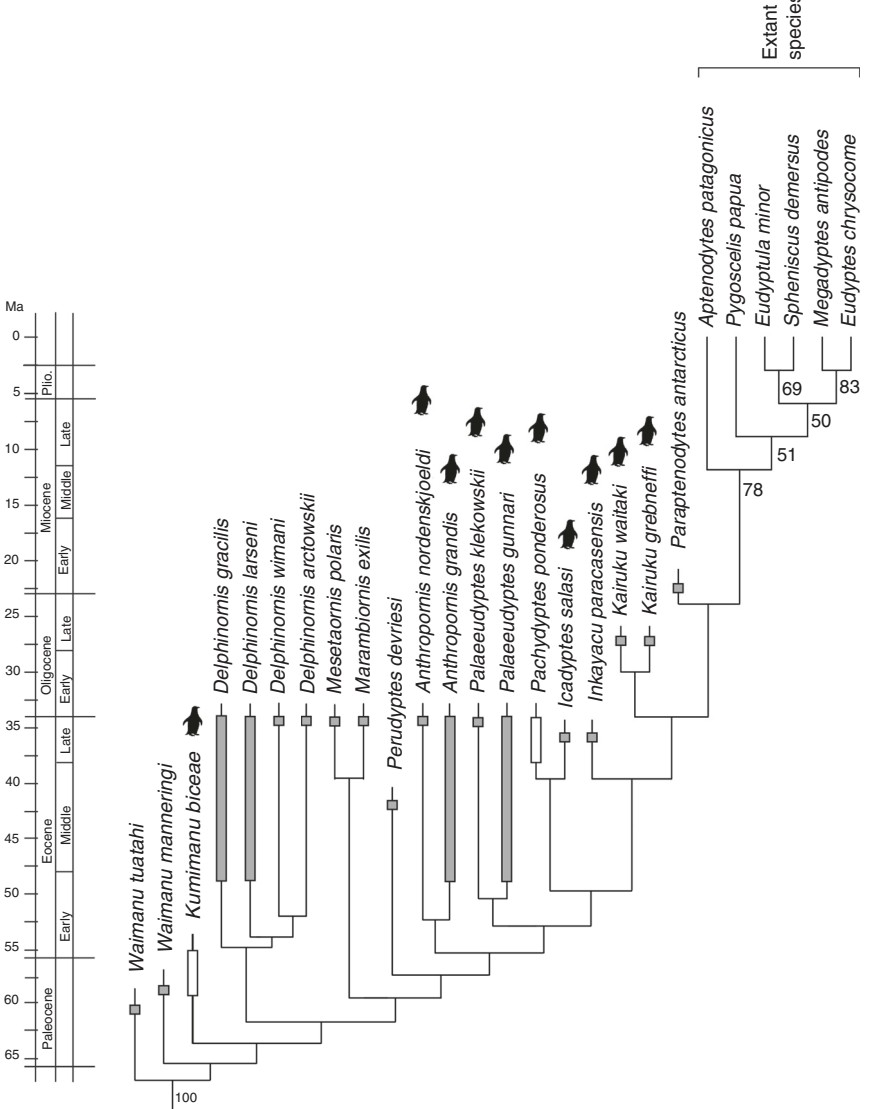

**Fig. 3** Phylogenetic interrelationships of the new penguin species. The phylogeny is based on the single most parsimonious tree resulting from the analysis of the reduced and emended data set ($L = 369$, $CI = 0.79$, $RI = 0.79$). The bars indicate the stratigraphic occurrence of the fossil species[2] (white bars denote an uncertain exact occurrence within the indicated range); internode length is hypothetical. Bootstrap support values (majority rule consensus) are indicated next to the internodes. The penguin icons denote giant taxa with a size exceeding that of the Emperor Penguin

other Paleocene penguins (Table 1) and would have had an unusually slender shaft relative to the massive proximal end of the bone, so we consider our longer length estimate to be more likely.

The fragment of the proximal end of the ulna exhibits half of the ventral cotyla, next to which there is a distinct, flange-like projection (Fig. 1d–f). The ulna of *W. tuatahi* and that of a phylogenetically basal penguin from the Waipara Greensand (CM 2016.6.1), whose description is pending[24], has two such flanges on the caudal and cranial surfaces of the bones, respectively: the olecranon and the dorsal cotylar process (Fig. 1g, h). Based on comparisons with the ulnae of the Sphenisciformes from the Waipara Greensand, we consider the flange on the ulna of the *K. biceae* holotype to be the dorsal cotylar process, because the olecranon has a more convex margin in Paleocene Sphenisciformes and is much more prominent in post-Paleocene taxa. A well-developed dorsal cotylar process is a plesiomorphic feature[24], which is absent in post-Paleocene Sphenisciformes.

The coracoid lacks the sternal end (Fig. 1b) and the acrocoracoid process is damaged, with its medial portion being broken. The scapular cotyla is of circular outline and more deeply excavated than in post-Paleocene Sphenisciformes including *Pachydyptes ponderosus*. The well-developed procoracoid process projects perpendicular to the longitudinal axis of the bone and has a ventrally deflected tip. As in *W. tuatahi* (Fig. 1c), there is no coracoidal fenestra, with this fenestra being present in all Eocene and Oligocene Sphenisciformes of which well-preserved coracoids are known (in *P. ponderosus*, the corresponding area is damaged, although the remaining medial section of the shaft suggests the former presence of a fenestra).

The cranial extremity of the scapula (Fig. 2a, b) resembles that of *Waimanu tuatahi* (Fig. 2c) and differs from the known scapulae of post-Paleocene penguins in that the humeral articulation facet is craniocaudally more extensive. Unlike in *W. tuatahi*, however, the scapular blade widens caudally, as it does in phylogenetically more crownward Sphenisciformes, and the

articulation facet of the humerus is craniocaudally narrower, with a more pointed ventral tip.

Of the sternum, the cranial margins of the carina and the corpus are preserved (Fig. 2g), but only a few osteological details can be discerned. As in *Platydyptes marplesi*[8] but unlike in extant penguins, the articular sulci for the coracoid meet each other at the midline of the bone. Compared to extant Sphenisciformes, the external lip of the articulation sulcus for the coracoid (labrum externum) is furthermore situated in a very lateral position. The sternal keel appears to have been unusually thick, but is too incompletely preserved to determine its original shape.

A thoracic vertebra (Fig. 2d, e) has a slightly concave caudal articulation facet and an essentially flat cranial one, whereas in extant Sphenisciformes, the vertebrae are opisthocoelous, that is, with a distinctly convex cranial articulation facet and a distinctly concave caudal one. Unlike in *W. tuatahi*, the lateral surfaces of the corpus do not exhibit well-delimited fossae.

The new species has a longer femur than all other fossil penguins, for which this skeletal element is known (Table 1). The bone is partially embedded in matrix (Fig. 2f) and is stouter than the femur of *W. tuatahi*, but less stout than the femora of *Kairuku* and *Inkayacu*. On the distal end, the fibular trochlea is strongly developed and there is a projecting tubercle for the gastrocnemialis lateralis muscle.

The tibiotarsus lacks the proximal end, but preserves the distalmost portion of the fibular crest (Fig. 2h). The distal end exhibits a crack, which disrupted the medial part of the bone, so that the condylus medialis appears to be unusually splayed. However, even when the bone is digitally restored (Fig. 2i), the shape of the distal end is highly characteristic in that the condyles are more widely spaced and proximodistally lower than in all other stem group Sphenisciformes, of which the tibiotarsus is known. The distal opening of the extensor sulcus is more medially positioned than in post-Paleocene Sphenisciformes, in which it is in a central position along the midline of the long axis of the tibiotarsus, but it is situated less medially than in *Waimanu manneringi* (Fig. 2j). The width of the distal end of the tibiotarsus is only exceeded by the very large Antarctic *P. klekowskii* specimens[14] (these consist of a humerus fragment and a tarsometatarsus, but an estimate of the distal width of the tibiotarsus is possible, because this value corresponds to the proximal width of the tarsometatarsus; Table 1).

**Results of the phylogenetic analysis.** Our initial phylogenetic analysis based on the complete data set of ref. [8] resulted in a placement of *Kumimanu* in a clade including all other Sphenisciformes except *W. manneringi* and *W. tuatahi*, where it was recovered in a polytomy together with the much smaller Eocene taxa *Perudyptes*, *Delphinornis*, *Marambiornis*, and *Mesetaornis* (Supplementary Fig. 1). Analysis of the reduced and emended data set (Methods section) supports a position of *K. biceae* outside a clade including all post-Paleocene Sphenisciformes, and recovers the new species as sister taxon of a clade including all other Sphenisciformes except *W. manneringi* and *W. tuatahi* (Fig. 3).

**Discussion**
Determining the size of extinct penguins from fragmentary remains is not straightforward as different species may have different proportions[8]. That *Kumimanu biceae*, n. gen. et sp. was an exceptionally large bird is, however, clear from the fact that the major limb bones are distinctly larger than those of most known Sphenisciformes (Table 1). Even our conservative estimate of 185 mm for the minimum total length of the humerus of *K. biceae* exceeds the humerus length of almost all other giant penguin species. The maximum estimate of 228 mm, which is based on the

humerus proportions of other Paleocene Sphenisciformes, is only surpassed by an estimate based on very large fossils from the late Eocene of Antarctica, which were assigned to *Palaeeudyptes klekowskii*[14] (Table 1).

Based on published allometric equations (Table 1 in ref. [11]) and a femur length of 161 mm, we calculate a weight of 101 kg and a body length of 1.77 m for *K. biceae*. These values exceed the weight and length estimates of all other stem group Sphenisciformes except for the exceptionally large specimens of *P. klekowskii*, for which a body length of about 2.0 m was estimated[14]. There exists no overlap between the bones of the holotype of *K. biceae* and those of the recently described large penguin foot bones from the Waipara Greensand (CM 2016.158.19)[4]. However, the proximal end of the tarsometatarsus of the latter species has a width of about 40.5 mm, and with a width of about 46 mm for the restored distal end of the tibiotarsus (Fig. 2j), *K. biceae* represents a larger bird. We therefore conclude that *K. biceae* is among the largest fossil penguins reported so far and may have been exceeded only in size by the above-mentioned fragmentary *P. klekowskii* specimens from the Eocene of Antarctica[14].

*K. biceae* differs from post-Paleocene Sphenisciformes in the plesiomorphic presence of a flange-like dorsal cotylar process of the ulna and in the plesiomorphic absence of a coracoidal fenestra. As in *W. tuatahi* but unlike in *C. unienwillia* and all post-Paleocene Sphenisciformes, the humerus of *K. biceae* has a slender shaft. The widening of the scapular blade, the more centrally positioned extensor sulcus of the tibiotarsus, and the absence of distinct fossae on the lateral surfaces of the thoracic vertebrae supports a closer relationship of *K. biceae* to phylogenetically more crownward post-Paleocene Sphenisciformes than to *W. manneringi* and *W. tuatahi*.

The tarsometatarsus of the unnamed giant penguin from the Waipara Greensand (CM 2016.158.1) has a more derived morphology than the tarsometatarsus of *Waimanu manneringi* and *W. tuatahi*[4], but as the tarsometatarsus is unknown for *K. biceae*, the affinities between the latter species and CM 2016.158.1 can not be assessed. The exact relationships between *K. biceae* and the slightly younger *Crossvallia unienwillia* are also uncertain due to the poor preservation of the holotype of the latter species. However, *K. biceae* was much larger than *C. unienwillia* and had a proportionally narrower humerus shaft (Fig. 1k–m).

Irrespective of the exact affinities between the new species and other Paleocene stem group Sphenisciformes, our study corroborates the observation of earlier authors[4, 6, 11] that some spenisciform species achieved a large size very early in the evolution of penguins. What is unexpected, however, is how fast these birds reached even their maximum size in the earliest Paleogene. That a penguin rivaling the largest previously known fossil species existed in the Paleocene may indicate that gigantism in penguins arose shortly after these birds became flightless divers. Gigantism therefore may be an inherent feature of Paleogene penguins, which may have evolved soon after aerodynamic constraints ceased to exist.

*Kumimanu* furthermore documents that a very large size occurred in a phylogenetically basal penguin, which in its osteological features closely resembles the Sphenisciformes from the Waipara Greensand. Because the new taxon is phylogenetically separated from the giant Eocene and Oligocene species by various smaller taxa, such as *Perudyptes*, *Delphinornis*, *Mesetaornis*, and *Marambiornis* (Fig. 3), a giant size must have evolved independently several times in penguin evolution.

There exists no correlation between palaeotemperatures and body size in fossil Sphenisciformes, with most giant taxa having evolved in a 'greenhouse world'[6, 18]. The presence of giant penguins in the Paleocene shortly after the end-Cretaceous mass extinction event may indicate that these birds entered ecological

niches left vacant after the extinction of large predatory marine reptiles[2]. An increase in body size over time is known for many groups of organisms, with this evolutionary trend ('Cope's Rule') being due to positive, directional selection on population level, which benefits survival and mating success[25]. In penguins, a size increase may have also constituted a selective advantage in competition for suitable breeding grounds. A correlation with improved diving capabilities has likewise been discussed[2].

To understand size evolution in penguins we therefore must not seek an explanation for the presence of giant species throughout the Paleogene, but one for the absence of equally large species in the Neogene and today. Earlier authors proposed that feeding competition with marine mammals played a role in the extinction of giant penguins and other very large wing-propelled diving birds[7, 26, 27], and competition with gregarious pinnipeds for safe breeding was also considered[28]. The disappearance of giant penguins indeed coincides with the rise of marine mammals, that is, odontocete cetaceans and pinnipeds[27], but the exact causes and mechanisms of a competitive replacement remain poorly understood. In any case, the evolution of penguins appears to have been strongly influenced by non-avian vertebrates: whereas the end-Cretaceous extinction of larger marine and terrestrial predators may have been the ecological driver for the loss of flight capabilities in the earliest Sphenisciformes, the radiation of pinnipeds and odontocetes towards the Oligocene[27] seems to have terminated the existence of giant species towards the Miocene.

## Methods

**Institutional repositories**. The fossils referred to here are in the collections of Canterbury Museum, Christchurch, New Zealand (CM); Museo de La Plata, La Plata, Argentina (MLP); Museum of New Zealand Te Papa Tongarewa, Wellington, New Zealand (NMNZ).

**Phylogenetic analysis**. The phylogenetic analysis is based on a published morphological character matrix (245 characters, without nuclear sequence data)[8]. Character scoring for the new fossil is as follows: 122:1, 123:0, 129:0, 131:0, 136:1, 137:1, 139:0, 140:2, 141:0, 149:0, 150:0, 151:1, 154:2, 155:0, 156:1, 157:1, 159:0, 160:1, 161:0, 196:1, 199:1, 235:1; all other characters were scored as unknown (?). Based on our study of the published fossils and various new specimens[24], character 140 (coracoidal fenestra) was scored as "2" (absent) for *Waimanu tuatahi*; the character was scored as "0" (complete) for *Anthropornis nordenskjoeldi*[29]. Character 141 (foramen nervi supracoracoidei) was scored as "0" (absent) for *W. tuatahi* and *A. nordenskjoeldi*. Based on the referred tibiotarsi in ref. [10], character 199 (position of sulcus extensorius) was scored as "1" (close to midline of tibiotarsus) for *Mesetaornis exilis*, *Marambiornis polaris*, *Delphinornis larseni*, *D. arctowskii*, *D. gracilis*, *D. wimani*, *Palaeeudyptes klekowskii*, and *P. gunnari*. The nexus file used in the analysis can be found in the Supplementary Data 1. In a second analysis (Supplementary Data 2), we deleted most non-sphenisciform species from the data set and left only *Phoebetria palpebrata* (Diomedeidae, Procellariiformes) as an outgroup taxon. To improve the resolution of the resulting phylogeny, we furthermore restricted the coverage of crown group Sphenisciformes to six species and deleted fossil taxa, which are poorly represented or irrelevant for the present study (*Palaeeudyptes antarcticus*, *Anthropornis* sp. [UCMP 321023], *Palaeospheniscus* spp., *Platydyptes* spp., *Duntroonornis*, *Archaeospheniscus* spp., *Eretiscus*, *Marplesornis*, *Madrynornis*). For this second analysis, two characters were newly added: character 246 (dorsal supracondylar process of ulna) was scored as present (0) for *P. palpebrata*, *W. tuatahi* and the new taxon, absent (1) for *Delphinornis larseni* (based on a referred ulna in ref. [10]), *Anthropornis nordenskjoeldi*, *Palaeeudyptes klekowskii*, *Kairuku grebneffi*, *K. waitaki*, *Inkayacu paracasensis*, *Icadyptes salasi*, and all extant sphenisciform species, and unknown (?) for all other species included in the analysis; character 247 (ratio of width of proximal end of humerus to width of shaft) was scored as 2.4 or more (0) for *P. palpebrata*, *W. tuatahi* and the new taxon, unknown (?) for *Mesetaornis polaris*, *Marambiornis exilis*, and *Delphinornis* spp., and < 2.4 (1) for all other species included in the analysis. All analyses were performed with the heuristic search modus of NONA 2.0[30] through the WIN-CLADA 1.00.08 interface[31], using the commands hold 10,000, mult*1000, hold/10, and max*. Character ordering was as in ref. [8]. Bootstrap support values were calculated with 200 replicates, ten searches holding ten trees per replicate, and TBR branch swapping without max*.

**Nomenclatural acts**. This published work and the nomenclatural acts it contains have been registered in ZooBank, the online registration system for the

International Code of Zoological Nomenclature. The ZooBank Life Science Identifiers for the taxa in this publication are http://zoobank.org/urn:lsid:zoobank.org: act:3EF7C725-A115-4A80-8FA8-2461342AEDEE and http://zoobank.org/urn:lsid: zoo-bank.org:act:596D5C8A-052A-48A4-81E2-67F0183BBAAA.

**Data availability**. The nexus files used in the phylogenetic analysis are available in the Supplementary Data 1 and 2. All other datasets generated during and/or analyzed during the current study are available from the corresponding author on reasonable request.

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

## Acknowledgements

We are indebted to Al Mannering for the preparation of the penguin fossil. Chris Clowes (GNS Science) provided data on the dinoflagellate fauna associated with the fossil. Jean-Claude Stahl (NMNZ) took the overview photos of the fossil before final preparation. We furthermore thank Carolina Acosta Hospitaleche, Martin de los Reyes, and Nadia Haidr (all MLP), and Ewan Fordyce and Marcus Richards (both Otago University) for access to fossil specimens. This research was supported by a grant from the Marsden Fund Council from New Zealand Government funding, managed by Royal Society Te Apārangi; the publication costs were covered by the RS Allan Fund.

## Author contributions

A.J.D.T. and R.P.S. collected the fossil. G.M. and A.J.D.T. designed the study. G.M., A.J. D.T., R.P.S., and V.L.D.P. collected data, made osteological comparisons and wrote the paper. All authors gave final approval for publication.

## Additional information

**Competing interests:** The authors declare no competing financial interests.

