## [Peer Review File · Nature Communications]

Reviewers' comments:

Reviewer #1 (Remarks to the Author):

A new specimen from the Paleocene of New Zealand is reported in this contribution. This finding is important in itself due to penguins from that age are very scarce. Even when the discussion about the origin of giant penguins is not new and it has been undertaken in previous publications, the available evidence is completely different from that of one decade ago. The analysis performed and the discussion of the results are completely appropriate, although I consider that data need to be previously adjusted.

The main problem I found are the data from specimens without a reliable systematic assignment. Some comments about this issue are made in the annotated manuscript. Once it has been remedied, the manuscript would be ready to be published.

Dra. Carolina Acosta Hospitaleche

Reviewer #2 (Remarks to the Author):

This paper describes one of the largest known penguins. Though the material is somewhat incomplete, the large size of the specimen is clear, and it would have rivaled the largest previously described penguins. This would be of only modest interest in and of itself, but is significant when considered in the context of the age of the specimen. Because the specimen demonstrates a very early date for gigantic size in penguins, it is noteworthy and deserving of publication.

My primary concerns regard the morphological evidence for the phylogenetic placement of the taxon. The placement one node above Waimanu is likely correct, but there are some issues that should be resolved:

1. One of the most important characters bearing on the placement of Kumimanu is the presence of *processus cotylaris dorsalis*. In the procellariiform *Calonectris*, the *processus cotylaris dorsalis* projects from the dorsal margin of the *cotyla dorsalis* (I assume the same is true of *Phoebetria*, though a specimen was not available). In CM 2016.6.1, the process identified as the *processus cotylaris dorsalis* appears to be placed at the border between the *cotyla dorsalis* and *cotyla ventralis* (or perhaps even from the *processus cotylaris ventralis*, which is not labeled), which is far displaced from the position in Procellariiformes and other birds. Thus I wonder if the structure in the penguin is homologous to the *processus cotylaris dorsalis*? This is important to consider, because if the feature is homologous to that in Procellariiformes it is primitive, but if it is not it may be a derived feature shared with Waimanu.
2. The supplementary matrix file contains two discrepancies: the name "Muriwaimanu tuatahi" for Waimanu tuatahi and the taxon *Spheniscus megaramphus* is included in the file but is not depicted in the phylogenetic tree and appears to have been excluded in the analysis discussed in the paper.

3. While I was able to replicate the phylogenetic results by excluding *Spheniscus megaramphus*, it should be noted that including this taxon results in 7 MPTs (with 0-length branches collapsed, 351 if such branches are not collapsed) and many more collapsed nodes.
4. The fact that including *Spheniscus megaramphus*, well-established as a stem representative of *Spheniscus*, causes many nodes to collapse highlights the fact that many taxa have been excluded apparently with the aim of increasing overall resolution. This may improve the "look" of the figure, but it likely introduces some nodes that are recovered only as a byproduct of sampling. I would suggest that the authors run an analysis with the full taxon sample to demonstrate the placement of *Kumimanu* stands (it does not need to be figured).
5. Likewise, it is unclear why the authors decided to exclude all soft tissue characters. Indeed, running the analysis with these characters re-instated resulted in a tree identical to the one presented in the paper. If the authors have a justification for excluding these characters, it should be stated in the text.
6. Regarding character 131 (spacing of coracoidal articulation sulci), I wonder if the sulci actually approach one another closely at midline in *Kumimanu*? It looks almost as if they overlap in the photo based on the thickness of that area. It is possible they both closely approach at midline and extend far to the lateral margins, as stem penguins tend to have a very wide sternal end of the coracoid compared to modern penguins.
7. Regarding character 140 (coracoidal fenestra), I am not certain this structure should be coded absent in *Waimanu* and *Kumimanu*. As originally formulated the presence of a proximal flange from the processus procoracoideus is considered evidence of presence of an incomplete coracoidal fenestra (see figure 21 of Bertelli and Giannini, 2005). This seems to be present in the figured coracoid of *Waimanu*. It is more difficult to decipher in *Kumimanu*, where only a very small proximal projection seems to be present but I cannot tell if there has been any damage.
8. Regarding character 199, The distal opening of the sulcus of *Kumimanu* appears to be almost exactly at the midline in figure 2 (i). It does appear to be placed medially in (h), but this seems to be due to damage.

Minor notes:

108: The reference for character ordering seems to be incorrect.

227: The reference for the *Palaeudyptes klekowskii* assignment seems to be incorrect.

Table 1: Because there is a substantial variation in limb bone length for many species (e.g., Jadwiszczak, 2006), so I would recommend adding a column to indicate the catalog number of the specimen measured for each species. I would also suggest adding a column for the ratio of the humerus proximal width to shaft width, since this is considered an important phylogenetic character in the paper.

Figure 1: In order to ease comparisons, I would recommend placing all of the *Kumimanu* humeri images in a row on the bottom half of the figure, with the *Crossvalia* and *Pachydyptes* to the right of (e) (h-m could then be moved up). This would make both morphological comparisons and the relative sizes of the humeri more clear.

Figure 2: Many of the abbreviations seem to be based on Latin terminology whereas the labels use English terms (e.g. car = sternal keel rather than carina and sac = coracoidal

articulation sulcus rather than sulcus articularis sternalis). Either is fine, but they should match.

Reviewer #3 (Remarks to the Author):

Comments by R Ewan Fordyce

KEY RESULTS: Please summarise what you consider to be the outstanding features of the work.

- A rare associated partial skeleton represents a new genus, Kumimanu, of basal penguin
- Age presumably late Paleocene, early in penguin history/phylogeny
- Major elements - although incomplete - extrapolate to large body size, similar to large penguins of late Eocene/Oligocene
- Suggests repeated convergent evolution of large body size

VALIDITY: Does the manuscript have flaws which should prohibit its publication? If so, please provide details.

No, there are no obvious flaws.

ORIGINALITY AND SIGNIFICANCE: If the conclusions are not original, please provide relevant references.

The material and conclusions are original.

On a more subjective note, do you feel that the results presented are of immediate interest to many people in your own discipline, and/or to people from several disciplines?

Yes, the results will be of immediate interest in the discipline; Kumimanu is new, and its phylogenetic position is shown clearly. Enough facts are given, and extrapolation explained, that those in the discipline will be able to judge size estimates for themselves. The proposed large body size in early penguin history should interest a wider audience.

DATA & METHODOLOGY: Please comment on the validity of the approach, quality of the data and quality of presentation. Please note that we expect our reviewers to review all data, including any extended data and supplementary information.

The textural data are descriptive, as normal for a partial skeleton of a single individual fossil vertebrate. It is fine to include comparisons and interpretations in the description. The terminology and level of detail here are expected and appropriate.

The material represents a large penguin. The skeleton is partial, and most of the bones present are incomplete. Nevertheless, the humerus and femur are comparable with, or perhaps larger than, any penguin species described to date.

Figures: because the core of the paper is inferred body size, the figures should be changed to show all key elements at the same scale and orientation, so that size is easily seen at a glance. This is particularly so on Fig. 1, for the incomplete humerus of the new penguin as compared with humeri from other named species: show all the humeri together, same scale, same orientation. (Consider also including humerus of *Kairuku grebneffi* referred, from Ksepka et al., which would be available from the Geology Museum at Univ Otago upon request.)

Figures: Fig. 2a is not needed; the digitally edited Fig. 2b is fine. Use same scale for Figs 2b and 2c, to better indicate size difference.

The Ksepka et al. 2012 phylogenetic matrix is highly suitable for this ms. However, there was a numbering error in the original 2012 published matrix, as explained on Dryad, from which the corrected file is available:

Ksepka DT, Fordyce RE, Ando T, Jones CM (2012) Data from: New fossil penguins (Aves, Sphenisciformes) from the Oligocene of New Zealand reveal the skeletal plan of stem penguins. Dryad Digital Repository. <http://dx.doi.org/10.5061/dryad.93j174jd.2>

The corrected file should be used.

Fig. 3 caption states "Strict consensus tree of the single most parsimonious tree" Presumably you do not mean single, but some number of equally parsimonious trees – specify the number.

I looked for a tree file (.tre) in the documents but could not see. But, I found the mix of data files and zip files confusing and might have missed it. If not already done, you should include the tree/s in a separate .tre file or in the nexus file.

Is the reporting of data and methodology sufficiently detailed and transparent to enable reproducing the results?

Yes.

APPROPRIATE USE OF STATISTICS AND TREATMENT OF UNCERTAINTIES: All error bars should be defined in the corresponding figure legends; please comment if that's not the case. Please include in your report a specific comment on the appropriateness of any statistical tests, and the accuracy of the description of any error bars and probability values.

Not applicable.

CONCLUSIONS: Do you find that the conclusions and data interpretation are robust, valid and reliable?

The conclusion of large penguin, toward the base of the sphenisciform clade, is justified on the basis of bone sizes and proportions, and phylogenetic analysis. The caution here relates to actual size and mass, as extrapolated from the femur.

As the authors note, skeletal proportions, and individual bone proportions vary amongst archaic penguins, even closely related species (e.g. as Ksepka et al. 2012 noted for the femur in *Kairuku* species). Body proportions are uncertain for most penguins basal to *Kairuku* – meaning most Paleocene-Eocene penguins - including, here, *Kumimanu*. Thus, size estimates based on single bones, in this case a femur (with support from an incomplete humerus), are venturesome extrapolations. But, this is the best that can be done for now.

Re: “That a penguin rivalling the largest previously known fossil species existed in the Paleocene demonstrates that gigantism in penguins arose shortly after these birds became flightless divers”

When do you think that flightless diving evolved? Before or after the K/Pg boundary?

SUGGESTED IMPROVEMENTS: Please list additional experiments or data that could help strengthening the work in a revision.

Geological age is stated as Teurian Stage (based on dinoflagellate dating), which is consistent with previous reports on the sequence. The Teurian is 10 Ma long, so better resolution is desirable. Morgans 2009 commented on the geology of the Moeraki Formation, whence came the penguin: “Moeraki Formation is assigned to the latest Paleocene (Teurian) NZP5”. Crouch & Brinkhuis 2005 showed dinoflagellate stratigraphy for Moeraki Formation, confirming zone NZP5, and including horizons potentially able to be matched with the penguin – see their Fig 7 which shows zone NZP5 (Crouch, E.M., Brinkhuis, H., 2005. Environmental change across the Paleocene–Eocene transition from eastern New Zealand... *Marine Micropaleontology* 56, 138–160). Crouch & Brinkhuis 2005 also show correlations for NZP5 (their Fig. 1: late Teurian, in range 55.5 – 59.5 Ma (old dates from 2005; need to check new absolute timescale).

Editing for style could shorten the ms.

REFERENCES: Does this manuscript reference previous literature appropriately? If not, what references should be included or excluded?

Yes, literature is ok, but above note that Crouch & Brinkhuis 2005 is important – include.

CLARITY AND CONTEXT: Is the abstract clear, accessible? Are abstract, introduction and conclusions appropriate?

The abstract could be shortened. Suggestions follow - -

Abstract A new giant fossil penguin (late Paleocene, New Zealand) is bigger than all other comparable species, and indicates the very early evolution of large body size in penguins. Several plesiomorphic features place *Kumimanu biceae*, n. gen et sp. basal to all post-Paleocene giant penguins. *Kumimanu* is phylogenetically separated from giant Eocene and Oligocene penguin species by various smaller taxa, which indicates that giant size evolved at least twice in penguin history. Giant penguins existed throughout most of the Paleogene and a marked size increase appears to be an intrinsic feature of Paleogene Sphenisciformes,

with the absence of very large species today most likely being due to the Oligo-Miocene radiation of marine mammals.

Please indicate any particular part of the manuscript, data, or analyses that you feel is outside the scope of your expertise, or that you were unable to assess fully.

I did not run the phylogenetic analysis to check the phylogeny, but from perusal of the .nex file I see no particular problems – providing that the tree file is included.

Reviewer #4 (Remarks to the Author):

The manuscript deals with a fossil penguin from the Paleocene of New Zealand that could be the largest penguin species ever described.

Authors

- 1) described the specimen as a new species and new genus,
- 2) performed a phylogenetic analysis to determine the phylogenetic position of the species,
- 3) estimated the original size of the skeletal elements and body,
- 4) discussed the rapid size increase in penguins, and
- 5) discussed the multiple origin of the gigantism in the evolutionary history in penguins.

The manuscript is soundly prepared, well-written and suitable for the publication. The importance of the fossil and the theme of the manuscript are not limited in the field of paleontology but have a strong influence to the field of the evolutionary biology and ecology too since the manuscript discusses the “natural experiment” about the body size increase and niche refilling under certain conditions. Please consider following minor to intermediate suggestions to improve the manuscript.

i) The description and the observation seems to tend to focus on the difference from Waimanu penguins but the presence of the plesiomorphic characters shared with Waimanu penguins is also important (i.e., in some skeletal elements, only the difference is described). It could be worthwhile to re-confirm characters shared between Waimanu and Kunimanu penguins when the theme of the manuscript is considered.

ii) I have no doubt that Kunimanu penguin is in basal position next to Waimanu penguins, but the relationship with other penguins seems not rigidly supported. Authors got single most parsimonious tree from rather simplified matrix based on previously published works. I am curious to know the result from non-simplified matrix and whether there is no possibility to support a clade of Waimanu and Kunimanu penguins after considering the issue i) if possible.

iii) Authors seem to emphasize the ‘multiple origin of gigantism in penguins’, but ‘the rapid size increase’ appears more important to me, and it is more rigorously presented in the manuscript. It can be said that the new specimen described here is merely another example to support the multiple origin of gigantism, not the new, definite evidence. If the ‘giant size’ and ‘phylogenetic separation’ are important to show the multiple origin, it seems that the result in Ksepka et al. (2012) already showed that: both *Kairuku* and *Anthorpornis* are

'giant' and they are phylogenetically separated. The result of the phylogenetic analysis based on the simplified matrix that excluded smaller-sized New Zealand penguins could be problematic in this context too. If the tendency of the size increase are found in mutually exclusive clades, it would be the definite evidence for the 'multiple origin of gigantism in penguins', i.e., if Waimanu and Kunimanu made a clade, that would be the evidence. It could be useful to use of the phylogram with geological ages between taxa to show that the Waimanu and Kunimanu are distant from all other penguins if not they make a clade. On the other hand, there is little room to doubt 'the rapid size increase' occurred in the Paleocene age, for Authors presented that Kunimanu is one of the largest, and possibly the largest fossil penguin and it is phylogenetically basal penguin next to Waimanu, the earliest and most basal penguins. The time that penguins required to achieve its maximum body size is revealed to be much shorter than the previous assumption: it must have taken long time from the early Paleocene to the middle to late Eocene. It could change our view on the body size increase as an evolutionary phenomenon.

Line 24: Some long noun clauses in the manuscript can be rewritten for better readability.

Line 27: 'intrinsic' should be rephrased. 'unique'?

Line 34: 'some' is somewhat conservative expression considering the contribution of the New Zealand penguins to the penguin's evolutionary history.

Line 47: What about *Anthorpnornis nordenskjöldi*?

Line 52: Why is not the largest *Palaeodyptes klekowskii* specimen mentioned here but later part of the manuscript?

Line 118: From one individual?

Line 133: Do authors think that the size is diagnostic features? If so, you can state that Kunimanu is distinguished from Waimanu spp. by its size.

Lines 153-163: This part could go to the discussion section and could be more concise.

Line 180: Please provide the comparison with the coracoid in *Pachydyptes*.

Lines 187-190: Please state that cranial part of the scapula is very similar to that in Waimanu and different from that in other penguins.

Line 227: Who did indicate that the assignment was 'tentative'?

Line 228: The body mass estimation is tricky business and theoretically an estimation based on extrapolation is not reliable so much. However, we do not have many choices to estimate the body mass of the extinct fossil penguins. It varies much depending on the selection of the skeletal elements. In the body mass estimation based on the width of the humeral head (Ando 2007), *Pachydyptes* was the heaviest (~130kg). Kunimanu had the wider humeral head thus the heavier body mass, I suppose.

Lines 227 and 230: The reference number is 19, not 1?

Line 239 and 258: Is it essential to discuss the affinities with other Waipara Greensand penguins?

Line 284: Might need references (Ando, 2007 or Ksepka and Ando, 2011).

Lines 282-287: What do authors want to convey? Niche issue is not directly related to the Southern inhabitation of penguins, and ptoptoids did not emerge in the Paleocene. It is true that in previous assumption, the size increase in penguins took long time. But if authors discuss this relating to the niche issue, please provide references or data that show the size increase in penguins is relatively longer compared to other marine organisms.

Line 288-292: Other hypothesis is the advantage in diving (Ando, 2007 or Ksepka and

Ando, 2011)

Line 301-303: Current data indicate that there was no such predatory odontocete nor pinnipeds in the Late Oligocene. It is possible to assume that there was a 'killer sperm whale' (Livyatan)-like odontocete existed in that age, but it is too speculative. As for pinnipeds, they have never achieved a giant form that could consume giant penguins and they just began to emerge in the Late Oligocene. Basiosauridae was possible predators for the Eocene giant penguins, but their diversity pattern was similar to that of penguins in the Eocene and unlikely to have consumed giant penguins to the extinction.

Line 348: There is no reference number 9.

Reviewer #1:

A new specimen from the Paleocene of New Zealand is reported in this contribution. This finding is important in itself due to penguins from that age are very scarce. Even when the discussion about the origin of giant penguins is not new and it has been undertaken in previous publications, the available evidence is completely different from that of one decade ago. The analysis performed and the discussion of the results are completely appropriate, although I consider that data need to be previously adjusted.

The main problem I found are the data from specimens without a reliable systematic assignment. Some comments about this issue are made in the annotated manuscript. Once it has been remedied, the manuscript would be ready to be published.

Response: We have modified the text according to the comments of the reviewer. In particular, we have now explicitly stated that the *Palaeudyptes klekowskii* material also includes more substantial specimens and added the references suggested by the reviewer.

We are aware of the fact that referral of isolated bones from Seymour Island can only be tentative. However, we followed Jadwiszczak (2006) in the referral of additional material to these species and now include this reference in the methods section. Because we are not aware of published criticism of Jadwiszczak (2006) that show his identifications to be erroneous, we consider our approach to be justified and hope that the reviewer agrees. Concerning the tibiotarsus, our approach is unproblematic, because all published tibiotarsi from the La Meseta Formation show the same condition concerning the scored character, and the same is true for the ulnae (an ulna was referred to *Delphinornis larseni* by Jadwiszczak).

Concerning the age of the Peruvian penguins: It is true that *Perudyptes* is from the middle Eocene Paracas Formation. The giant taxa *Inkayacu* and *Icadyptes*, however, are from the Otuma Formation, for which a late Eocene age of 36 million years is indicated by both Clarke et al. (2007) and Clarke et al. (2010).

Reviewer #2:

This paper describes one of the largest known penguins. Though the material is somewhat incomplete, the large size of the specimen is clear, and it would have rivaled the largest previously described penguins. This would be of only modest interest in and of itself, but is significant when considered in the context of the age of the specimen. Because the specimen demonstrates a very early date for gigantic size in penguins, it is noteworthy and deserving of publication.

My primary concerns regard the morphological evidence for the phylogenetic placement of the taxon. The placement one node above Waimanu is likely correct, but there are some issues that should be resolved:

1. One of the most important characters bearing on the placement of Kumimanu is the presence of processus cotylaris dorsalis. In the procellariiform *Calonectris*, the processus cotylaris dorsalis projects from the dorsal margin of the cotyla dorsalis (I assume the same is true of *Phoebetria*, though a specimen was not available). In CM 2016.6.1, the process identified as the processus cotylaris dorsalis appears to be placed at the border between the cotyla dorsalis and cotyla ventralis (or perhaps even from the processus cotylaris ventralis, which is not labeled), which is far displaced from the position in Procellariiformes and other birds. Thus I wonder if the structure in the penguin is homologous to the processus cotylaris dorsalis? This is important to consider, because if the feature is homologous to that in Procellariiformes it is primitive, but if it is not it may be a derived feature shared with Waimanu.

Response: We agree with the reviewer that the shape of the processus cotylaris dorsalis of Paleocene penguins is somewhat different from that in procellariiforms. However, the whole

proximal end of the ulna of Paleocene penguins is greatly modified compared to that of procellariiforms. Certainly the similarity in the shape of the processus cotylaris dorsalis of Paleocene penguins and procellariiforms is greater than that of the olecranon, whose homology has not been questioned (see attached picture below). We have a paper in press in the *Journal of Vertebrate Paleontology*, in which this issue is discussed in more detail, and have now added a reference to this study to the present manuscript. The JVP paper has just been accepted and should appear online in 1-2 months.

2. The supplementary matrix file contains two discrepancies: the name “Muriwaimanu tuatahi” for *Waimanu tuatahi* and the taxon *Spheniscus megaramphus* is included in the file but is not depicted in the phylogenetic tree and appears to have been excluded in the analysis discussed in the paper.

Response: This was a lapsus that is now fixed (see also comments below).

3. While I was able to replicate the phylogenetic results by excluding *Spheniscus megaramphus*, it should be noted that including this taxon results in 7 MPTs (with 0-length branches collapsed, 351 if such branches are not collapsed) and many more collapsed nodes.

Response: We have now reanalyzed the data with the full data set and added the phylogeny based on the full data set of Ksepka et al. (2012) as a supplementary figure (see also comments below).

4. The fact that including *Spheniscus megaramphus*, well-established as a stem representative of *Spheniscus*, causes many nodes to collapse highlights the fact that many taxa have been excluded apparently with the aim of increasing overall resolution. This may improve the “look” of the figure, but it likely introduces some nodes that are recovered only as a byproduct of sampling. I would suggest that the authors run an analysis with the full taxon sample to demonstrate the placement of *Kumimanu* stands (it does not need to be figured).

Response: Yes, we have indeed removed a number of fossil taxa from the analysis to improve the resolution of the tree. We have explicitly stated this in the previous version and our rationale for doing so was the poor representation of some of the fossils and the fact that many of these are highly unlikely to have bearing on the placement of Paleocene fossils (as they are positioned too “high” in the tree). However, we have also analyzed our data with the full data set of Ksepka et al. (2012) and found the results to be in concordance with our main conclusion. Following the suggestion of the reviewer, we have now included the results of this analysis as a supplementary figure in the paper (Supplementary figure 1).

5. Likewise, it is unclear why the authors decided to exclude all soft tissue characters. Indeed, running the analysis with these characters re-instated resulted in a tree identical to the one presented in the paper. If the authors have a justification for excluding these characters, it should be stated in the text.

Response: We excluded the soft tissue characters, because they are unknown from all fossils and therefore cannot bear any phylogenetic information for the placement of these taxa (the extant species are too deeply nested in the phylogeny for their interrelationships to be significant for the placement of the Paleocene taxa). However, following the suggestion of the reviewer, we have now maintained the soft tissue characters in the new analysis of the reduced data set.

6. Regarding character 131 (spacing of coracoidal articulation sulci), I wonder if the sulci actually approach one another closely at midline in Kumimanu? It looks almost as if they overlap in the photo based on the thickness of that area. It is possible they both closely approach at midline and extend far to the lateral margins, as stem penguins tend to have a very wide sternal end of the coracoid compared to modern penguins.

Response: This is correct and we have changed the character scoring in the analysis and added a note to the description.

7. Regarding character 140 (coracoidal fenestra), I am not certain this structure should be coded absent in Waimanu and Kumimanu. As originally formulated the presence of a proximal flange from the processus procoracoideus is considered evidence of presence of an incomplete coracoidal fenestra (see figure 21 of Bertelli and Giannini, 2005). This seems to be present in the figured coracoid of Waimanu. It is more difficult to decipher in Kumimanu, where only a very small proximal projection seems to be present but I cannot tell if there has been any damage.

Response: We have studied various unpublished specimens of Waimanu for a paper in press (Mayr et al. in press, now cited in the Kumimanu study) and these fossils clearly show that even an incomplete coracoidal fenestra is absent in Waimanu. We have now added a note about this in the manuscript.

8. Regarding character 199, The distal opening of the sulcus of Kumimanu appears to be almost exactly at the midline in figure 2 (i). It does appear to be placed medially in (h), but this seems to be due to damage.

Response: We agree and have changed the character scoring in the analysis.

Minor notes:

108: The reference for character ordering seems to be incorrect.

Response: fixed

227: The reference for the *Palaeudyptes klekowskii* assignment seems to be incorrect.

Response: fixed

Table 1: Because there is a substantial variation in limb bone length for many species (e.g., Jadwiszczak, 2006), so I would recommend adding a column to indicate the catalog number of the specimen measured for each species. I would also suggest adding a column for the ratio of the humerus proximal width to shaft width, since this is considered an important phylogenetic character in the paper.

Response: We have taken the measurements from publications that are indicated with superscript letters next to the values. In these publications, the specimen numbers are indicated, so that interested readers can trace the fossils. In two cases, our own measurements for *Anthropornis nordenskjoeldi* are given and the specimen number is indicated in the footnote. We hope that this is sufficient, because adding specimen numbers to all measurements would make the table very confusing. Because we have not measured most specimens ourselves but took values from the literature, it would be very circumstantial for us to provide measurements of the proximal humerus width, since the published measurements (if they do exist at all) are not always consistent (i.e., different landmarks are used for the measurement). We therefore hope that we can leave the table as it is.

Figure 1: In order to ease comparisons, I would recommend placing all of the Kumimanu humeri images in a row on the bottom half of the figure, with the *Crossvalia* and *Pachydyptes* to the right of

(e) (h-m could then be moved up). This would make both morphological comparisons and the relative sizes of the humeri more clear.

Response: We have changed the figure according to these suggestions and agree that it is clearer now

Figure 2: Many of the abbreviations seem to be based on Latin terminology whereas the labels use English terms (e.g. car = sternal keel rather than carina and sac = coracoidal articulation sulcus rather than sulcus articularis sternalis). Either is fine, but they should match.

Response: fixed

Reviewer #3:

Figures: because the core of the paper is inferred body size, the figures should be changed to show all key elements at the same scale and orientation, so that size is easily seen at a glance. This is particularly so on Fig. 1, for the incomplete humerus of the new penguin as compared with humeri from other named species: show all the humeri together, same scale, same orientation. (Consider also including humerus of *Kairuku grebneffi* referred, from Ksepka et al., which would be available from the Geology Museum at Univ Otago upon request.

Response: We have now modified Fig. 1, based on the comments of reviewer 2. The humeri were shown to scale in the previous version and are likewise shown to scale in the current version of the figure. We greatly appreciate the offer to include a figure of the *Kairuku* humerus, but would rather refrain from doing so as to maintain the large size of the figure.

Figures: Fig. 2a is not needed; the digitally edited Fig. 2b is fine. Use same scale for Figs 2b and 2c, to better indicate size difference.

Response: We have now shown Figs 2b and 2c to scale. However, if possible, we would like to leave Fig. 2a, so that readers can immediately see the original, unaltered photograph. Because the widening of the scapular shaft is one of the important feature we discuss, we consider it important to show both the edited and the unaltered photo.

The Ksepka et al. 2012 phylogenetic matrix is highly suitable for this ms. However, there was a numbering error in the original 2012 published matrix, as explained on Dryad, from which the corrected file is available:

Ksepka DT, Fordyce RE, Ando T, Jones CM (2012) Data from: New fossil penguins (Aves, Sphenisciformes) from the Oligocene of New Zealand reveal the skeletal plan of stem penguins.

Dryad Digital Repository. <http://dx.doi.org/10.5061/dryad.93j174jd.2>

The corrected file should be used.

Response: We have now reanalyzed the data with the corrected file on Dryad.

Fig. 3 caption states "Strict consensus tree of the single most parsimonious tree" Presumably you do not mean single, but some number of equally parsimonious trees – specify the number.

Response: This was a lapsus. Actually, only a single tree resulted from the analysis.

I looked for a tree file (.tre) in the documents but could not see. But, I found the mix of data files and zip files confusing and might have missed it. If not already done, you should include the tree/s in a separate .tre file or in the nexus file.

Response: We have now provided both, nexus and tree files.

Re: "That a penguin rivalling the largest previously known fossil species existed in the Paleocene demonstrates that gigantism in penguins arose shortly after these birds became flightless divers"

When do you think that flightless diving evolved? Before or after the K/Pg boundary?

Response: We cannot answer this question, but would nevertheless like to leave the statement in the abstract, especially as reviewer 4 considered this to be one of our major conclusions. We think that even if flightless penguins already evolved in the late Cretaceous, our statement would be true. However, to account for these uncertainties, we have rewritten the sentence and now say that our findings “may indicate that” (rather than “demonstrate”)

Geological age is stated as Teurian Stage (based on dinoflagellate dating), which is consistent with previous reports on the sequence. The Teurian is 10 Ma long, so better resolution is desirable. Morgans 2009 commented on the geology of the Moeraki Formation, whence came the penguin: “Moeraki Formation is assigned to the latest Paleocene (Teurian) NZP5”. Crouch & Brinkhuis 2005 showed dinoflagellate stratigraphy for Moeraki Formation, confirming zone NZP5, and including horizons potentially able to be matched with the penguin – see their Fig 7 which shows zone NZP5 (Crouch, E.M., Brinkhuis, H., 2005. Environmental change across the Paleocene–Eocene transition from eastern New Zealand... *Marine Micropaleontology* 56, 138–160). Crouch & Brinkhuis 2005 also show correlations for NZP5 (their Fig. 1: late Teurian, in range 55.5 – 59.5 Ma (old dates from 2005; need to check new absolute timescale).

Response: We have now included Crouch & Brinkhuis 2005

REFERENCES: Does this manuscript reference previous literature appropriately? If not, what references should be included or excluded?

Yes, literature is ok, but above note that Crouch & Brinkhuis 2005 is important – include.

Response: We have now included Crouch & Brinkhuis 2005

The abstract could be shortened. Suggestions follow - -

Abstract A new giant fossil penguin (late Paleocene, New Zealand) is bigger than all other comparable species, and indicates the very early evolution of large body size in penguins. Several plesiomorphic features place *Kumimanu biceae*, n. gen et sp. basal to all post-Paleocene giant penguins. *Kumimanu* is phylogenetically separated from giant Eocene and Oligocene penguin species by various smaller taxa, which indicates that giant size evolved at least twice in penguin history. Giant penguins existed throughout most of the Paleogene and a marked size increase appears to be an intrinsic feature of Paleogene Sphenisciformes, with the absence of very large species today most likely being due to the Oligo-Miocene radiation of marine mammals.

Response: Following these suggestions, we have now modified the abstract, although we would like to maintain some sentences from the original version. We hope that this is considered acceptable.

Reviewer #4:

i) The description and the observation seems to tend to focus on the difference from Waimanu penguins but the presence of the plesiomorphic characters shared with Waimanu penguins is also important (i.e., in some skeletal elements, only the difference is described). It could be worthwhile to re-confirm characters shared between Waimanu and *Kumimanu* penguins when the theme of the manuscript is considered.

Response: We are aware of the great overall similarities between the new fossil and the Waipara penguins, but consider it likely that these are due to retained plesiomorphic characteristics. We have a manuscript in press, in which new penguin material from the Waipara Greensand is described in detail, and have now added a reference to this study. We refrain from too detailed descriptions in the present study due to the Word limit of Nature Communications and the fact that a meaningful detailed comparison has to await the publication of the comprehensive JVP study (and is beyond the scope of the present study). We hope that this is an acceptable approach.

ii) I have no doubt that Kunimanu penguin is in basal position next to Waimanu penguins, but the relationship with other penguins seems not rigidly supported. Authors got single most parsimonious tree from rather simplified matrix based on previously published works. I am curious to know the result from non-simplified matrix and whether there is no possibility to support a clade of Waimanu and Kunimanu penguins after considering the issue i) if possible.

Response: We have now also added the results of the analysis of the full data set from the Ksepka et al. (2012) study as a supplementary figure, and the resulting phylogeny is in concordance with our placement of *Kumimanu*. Although the position of the new taxon is less well resolved with this data set, it is clearly placed outside the Waimanu clade.

iii) Authors seems to emphasize the 'multiple origin of gigantism in penguins', but 'the rapid size increase' appears more important to me, and it is more rigorously presented in the manuscript. It can be said that the new specimen described here is mere another example to support the multiple origin of gigantism, not the new, definite evidence. If the 'giant size' and 'phylogenetic separation' are important to show the multiple origin, it seems that the result in Ksepka et al. (2012) already showed that: both *Kairuku* and *Anthropornis* are 'giant' and they are phylogenetically separated. The result of the phylogenetic analysis based on the simplified matrix that excluded smaller-sized New Zealand penguins could be problematic in this context too. If the tendency of the size increase are found in mutually exclusive clades, it would be the definite evidence for the 'multiple origin of gigantism in penguins', i.e., if Waimanu and Kunimanu made a clade, that would be the evidence. It could be useful to use of the phylogram with geological ages between taxa to show that the Waimanu and Kunimanu are distant from all other penguins if not they make a clade.

On the other hand, there is little room to doubt 'the rapid size increase' occurred in the Paleocene age, for Authors presented that Kunimanu is one of the largest, and possibly the largest fossil penguin and it is phylogenetically basal penguin next to Waimanu, the earliest and most basal penguins. The time that penguins required to achieve its maximum body size is revealed to be much shorter than the previous assumption: it must have taken long time from the early Paleocene to the middle to late Eocene. It could change our view on the body size increase as an evolutionary phenomenon.

Response: We deliberately did not comment -too-much on the rapid size increase, because it is actually unknown exactly when penguins split from their sister taxon and lost their flight capabilities. In our view, this limits a meaningful discussion of the topic.

It is true that *Anthropornis* and *Kairuku* are separated in the Ksepka et al. analysis. However, the position of these two taxa in the Ksepka et al. study would also be in accordance with the assumption that a giant size evolved in the stem lineage of penguins and was later lost in the lineage leading to the crown group. From the results of this study it can therefore not be firmly concluded that "giant size" evolved more than once independently, especially as all taxa separating *Anthropornis* and *Kairuku* are very large (whereas the taxa separating *Kumimanu* and the later giant forms, such as *Delphinornis* and *Mesetaornis*, are quite small).

Based on these comments, we have now modified Fig. 3 and also show the stratigraphic occurrence of the taxa.

Line 24: Some long noun clauses in the manuscript can be rewritten for better readability.

Response: We have now tried to reduce these.

Line 27: 'intrinsic' should be rephrased. 'unique'?

Response: We have now substituted this with "inherent", as "unique" would not convey what we meant.

Line 34: 'some' is somewhat conservative expression considering the contribution of the New Zealand penguins to the penguin's evolutionary history.

Response: We have now substituted this with "considerably"

Line 47: What about *Anthropornis nordenskjöldi*?

Response: We have modified the introduction accordingly and now list both *Pachydyptes* and *Anthropornis* as the largest taxa.

Line 52: Why is not the largest *Palaeudyptes klekowskii* specimen mentioned here but later part of the manuscript?

Response: We have added a note on this specimen into the introduction.

Line 118: From one individual?

Response: Yes, we have added a note on this

Line 133: Do authors think that the size is diagnostic features? If so, you can state that *Kunimanu* is distinguished from *Waimanu* spp. by its size.

Response: We do not think that size is a diagnostic feature on genus-level and therefore did not include the large size in the diagnosis

Lines 153-163: This part could go to the discussion section and could be more concise.

Response: We have changed this according to this suggestion.

Line 180: Please provide the comparison with the coracoid in *Pachydyptes*.

Response: We have now added a few comments on the coracoid of *Pachydyptes*.

Lines 187-190: Please state that cranial part of the scapula is very similar to that in *Waimanu* and different from that in other penguins.

Response: We have now added a comment on this.

Line 227: Who did indicate that the assignment was 'tentative'?

Response: This was an error and we have deleted the word.

Line 228: The body mass estimation is tricky business and theoretically an estimation based on extrapolation is not reliable so much. However, we do not have many choices to estimate the body mass of the extinct fossil penguins. It varies much depending on the selection of the skeletal elements. In the body mass estimation based on the width of the humeral head (Ando 2007), *Pachydyptes* was the heaviest (~130kg). *Kunimanu* had the wider humeral head thus the heavier body mass, I suppose.

Response: We agree that calculating body masses from isolated bones is difficult and our estimates should be taken as rough only. Virtually all bones of the new taxon are, however, larger than those of previously known fossil penguins, so our main conclusions are well-supported.

Lines 227 and 230: The reference number is 19, not 1?

Response: fixed.

Line 239 and 258: Is it essential to discuss the affinities with other *Waipara Greensand* penguins?

Response: see above concerning the JVP paper in press.

Line 284: Might need references (Ando, 2007 or Ksepka and Ando, 2011).

Response: We have now added a reference to Ksepka and Ando, 2011

Lines 282-287: What do author want to convey? Niche issue is not directly related to the Southern inhabitation of penguins, and ptopteids did not emerge in the Paleocene. It is true that in previous assumption, the size increase in penguins took long time. But if authors discuss this relating to the niche issue, please provide references or data that show the size increase in penguins is relatively longer compared to other marine organisms.

Response: We have now deleted these sentences

Line 288-292: Other hypothesis is the advantage in diving (Ando, 2007 or Ksepka and Ando, 2011)

Response: We have now added a reference to Ksepka and Ando, 2011

Line 301-303: Current data indicate that there was no such predatory odontocete nor pinnipeds in the Late Oligocene. It is possible to assume that there was a 'killer sperm whale' (*Livyatan*)-like

odontocete existed in that age, but it is too speculative. As for pinnipeds, they have never achieved a giant form that could consume giant penguins and they just began to emerge in the Late Oligocene. Basiosauridae was possible predators for the Eocene giant penguins, but their diversity pattern was similar to that of penguins in the Eocene and unlikely to have consumed giant penguins to the extinction.

Response: We have now formulated this sentence more cautiously and just hint at a potential perspective for future studies.

Line 348: There is no reference number 9.

Response: fixed

Reviewers' comments:

Reviewer #1 (Remarks to the Author):

I do not have any different comment, I would only like to strengthen my main concern about the scoring made on non-associated elements preliminarily assigned by Jadwiszczak 2006 to the different Antarctic species without comparative elements.

Reviewer #2 (Remarks to the Author):

The revision address my concerns regarding the phylogeny and corrects all minor errors. There is one lingering issue. I am still not convinced that the ulna is correctly labeled. The authors included a figure in the rebuttal letter, and I believe there is still some confusion over the dorsal cotyla and ventral cotyla. I have attached an image of the right ulna of the procellariform *Calonectris diomedea* with these structures labeled according to Baumel and Witmer (1993). Unless I have made an error, or the ulna of *Diomedeoides* is very different from that of *Calonectris*, I believe the structure labeled dorsal cotyla in the figures of the procellariform and fossil penguin is instead the ventral cotyla, whereas the structure labeled dorsal cotylar process appears to be the dorsal cotyla itself in the image of the procellariform.

The may be good news, as if this is indeed an error, it would explain confusion over homology. If the structure labeled dorsal cotyla in the figure of the penguin is actually the ventral cotyla, I believe the authors are likely to be correct in homologizing this structure with that in procellariform (but was unable to understand this when the figure was mislabeled). If the authors agree with this assessment, I consider the payment ready for publication after the figure, text and supplement are lightly edited to correct this issue.

Reviewer #3 (Remarks to the Author):

Mayr et al Kumimanu ms

This version of the ms generally addresses earlier concerns. I don't have many comments.

Taxonomy has been clarified (in response to reviewer 1). Yes, Seymour Island material is a perennial problem, and yes it is dealt with satisfactorily.

On the issue of body size, the ms uses several different superlatives for "giant". The term giant is not defined, but there is a comment in supplementary files that implies giant to be larger than that of the Emperor penguin. This definition should appear early in the ms, preferably in the introduction.

Also, best decide on one superlative to apply to the bird and stick with it. On p 1 alone we are told that the bird is colossal, extremely large, and giant.

The interpretation of phylogenetics is clearer (in response to reviewer 2). Explanation of soft tissue characters is ok; it is reasonable to omit them in an analysis on stem penguins.

Some phylogenetic terminology could be addressed:

L31 and elsewhere: "stem group" can be shortened to stem without loss of meaning, e.g. ... Paleocene stem Sphenisciformes...

L33. The word stem should not be used to mean "to originate or arise from", because of confusion with the use of the term stem in phylogenetics.

L38. "unnamed species that is phylogenetically more derived..." "Derived" should be reserved for character states. Here, and elsewhere, it would be better to say more-crownward.

Fig 1 is improved by use of rescaled photos.

L 149. "... phylogenetically more advanced... is better stated as more-crownward.

Geological age is clearer (in response to reviewer 3). Cladistic issues are actioned. The abstract has been suitably revised.

Phylogeny issues are fixed (in response to reviewer 4). Predation-associated extinction is considered and referenced.

Reviewer #4 (Remarks to the Author):

Authors well-responded to my comments and questions and justified their view in an adequate manner. It would be a worthwhile contribution to the field of evolutionary biology.

Response to the reviewers:

Reviewer #1 (Remarks to the Author):

I do not have any different comment, I would only like to strengthen my main concern about the scoring made on non-associated elements preliminarily assigned by Jadwiszczak 2006 to the different Antarctic species without comparative elements.

- **Response:** we appreciate and understand these concerns, but as detailed in our previous response, we consider our approach to be justified by the published data.

Reviewer #2 (Remarks to the Author):

The revision address my concerns regarding the phylogeny and corrects all minor errors. There is one lingering issue. I am still not convinced that the ulna is correctly labeled. The authors included a figure in the rebuttal letter, and I believe there is still some confusion over the dorsal cotyla and ventral cotyla. I have attached an image of the right ulna of the procellariiform *Calonectris diomedea* with these structures labeled according to Baumel and Witmer (1993). Unless I have made an error, or the ulna of *Diomedeoides* is very different from that of *Calonectris*, I believe the structure labeled dorsal cotyla in the figures of the procellariiform and fossil penguin is instead the ventral cotyla, whereas the structure labeled dorsal cotylar process appears to be the dorsal cotyla itself in the image of the procellariiform.

This may be good news, as if this is indeed an error, it would explain confusion over homology. If the structure labeled dorsal cotyla in the figure of the penguin is actually the ventral cotyla, I believe the authors are likely to be correct in homologizing this structure with that in procellariiform (but was unable to understand this when the figure was mislabeled). If the authors agree with this assessment, I consider the paper ready for publication after the figure, text and supplement are lightly edited to correct this issue.

- **Response:** You are of course right and we have changed this embarrassing mistake (which did not bear on our interpretation of the data).

Reviewer #3 (Remarks to the Author):

On the issue of body size, the ms uses several different superlatives for "giant". The term giant is not defined, but there is a comment in supplementary files that implies giant to be larger than that of the Emperor penguin. This definition should appear early in the ms, preferably in the introduction.

- **Response:** fixed – we have now added such a note

Also, best decide on one superlative to apply to the bird and stick with it. On p 1 alone we are told that the bird is colossal, extremely large, and giant.

- **Response:** fixed

Some phylogenetic terminology could be addressed:

L31 and elsewhere: "stem group" can be shortened to stem without loss of meaning, e.g. ... Paleocene stem Sphenisciformes...

- **Response:** we have now substituted or deleted the term "stem group" in most cases. We left it in a few places, because "stem group" is the formally correct term, whereas "stem" is a more informal circumscription

L33. The word stem should not be used to mean “to originate or arise from”, because of confusion with the use of the term stem in phylogenetics.

- **Response:** fixed

L38. “unnamed species that is phylogenetically more derived...” “Derived” should be reserved for character states. Here, and elsewhere, it would be better to say more-crownward.

- **Response:** fixed

L 149. “... phylogenetically more advanced... is better stated as more-crownward.

- **Response:** fixed